# Self-Sensing Nanocomposites for Structural Applications: Choice Criteria

**DOI:** 10.3390/nano11040833

**Published:** 2021-03-24

**Authors:** Liberata Guadagno, Patrizia Lamberti, Vincenzo Tucci, Luigi Vertuccio

**Affiliations:** 1Department of Industrial Engineering, University of Salerno, Via Giovanni Paolo II, 84084 Fisciano, Italy; 2NANO_MATES, Research Centre for Nanomaterials and Nanotechnology at the University of Salerno, University of Salerno, Via Giovanni Paolo II, 132, 84084 Fisciano, Italy; plamberti@unisa.it (P.L.); vtucci@unisa.it (V.T.); 3Department of Information and Electrical Engineering and Applied Mathematics, University of Salerno, Via Giovanni Paolo II, 84084 Fisciano, Italy

**Keywords:** carbon nanoparticles, electrical percolation threshold, self-sensing, mechanical properties

## Abstract

Epoxy resins containing multi-wall carbon nanotubes (MWCNTs) have proven to be suitable for manufacturing promising self-sensing materials to be applied in the automotive and aeronautic sectors. Different parameters concerning morphological and mechanical properties of the hosting matrices have been analyzed to choose the most suitable system for targeted applications. Two different epoxy precursors, the tetrafunctional tetraglycidyl methylene dianiline (TGMDA) and the bifunctional bisphenol A diglycidyl ether (DGEBA) have been considered. Both precursors have been hardened using the same hardener in stoichiometric conditions. The different functionality of the precursor strongly affects the crosslinking density and, as a direct consequence, the electrical and mechanical behavior. The properties exhibited by the two different formulations can be taken into account in order to make the most appropriate choice with respect to the sensing performance. For practical applications, the choice of one formulation rather than another can be performed on the basis of costs, sensitivity, processing conditions, and most of all, mechanical requirements and in-service conditions of the final product. The performed characterization shows that the nanocomposite based on the TGMDA precursor manifests better performance in applications where high values in the glass transition temperature and storage modulus are required.

## 1. Introduction

Epoxy resins containing dispersed electrically conductive nanofillers have proven to be effective multifunctional materials since they are able to integrate, together with structural and thermal properties, functional abilities [1,2,3,4,5,6,7,8,9,10,11,12,13,14,15,16,17,18]. Among “conductive nanofillers”, unfunctionalized and functionalized CNTs have been extensively explored to impart functional properties to polymeric composite materials. In particular, many synthesis procedures have been developed to modify the nanofiller, which may serve to enhance the performance of specific properties in polymeric materials widening the range of their possible applications [19].

Carbon nanotubes (CNTs) are able to confer self-protection functions to polymeric/composite materials such as lightning strike protection [10], anticorrosive properties [20,21], resistance to UV irradiation [22,23]. They can also be used to integrate self-responsive functions in epoxy composites [5,14,15]. Anti/de-icing function can be imparted to unreinforced and carbon fibers reinforced composites (CFRCs) [24]. Furthermore, together with anti/de-icing, self-protection, etc., other functions such as self-healing, self-protection, self-sensing and damage monitoring can be imparted to the nanocomposites [5,24,25].

With regard to the sensing property, the change in electrical resistance has been found to be directly correlated to the variation of the applied tensile and compression stress [26,27]. Moreover, sensors based on epoxy resins filled with low percentages of carbon nanostructures have proven to be suitable also for monitoring the appearance of microcracks that may affect the integrity of structural parts [15,17]. The piezoresistive property of epoxy-based formulations has been extensively exploited in the literature for imparting self-sensing and health-monitoring function to structural resins [12,15,17]. However, the correlation of the self-sensing properties with the chemical structure of the hosting epoxy precursor has never been investigated in the literature due to the different epoxy precursors employed and, most of all, the different hardeners chosen by several authors [28,29]. This aspect, together with the correlation of the chemical nature of the precursor and the mechanical performance of the resulting nanocomposite is to be investigated for practical implications and for organizing a vademecum to follow in the choice of the materials to design lightweight sensors integrated into the structural components. Here, these aspects are analyzed for two different epoxy precursors, tetraglycidyl methylene dianiline (TGMDA) and bisphenol A diglycidyl ether (DGEBA) hardened in stoichiometric condition with the same hardener (4,4 diaminodiphenylsulfone). This choice allowed factoring in the different parameters and focusing the attention only on the correlations between the properties of the nanocomposites and the nature of the epoxy precursor. For the TGMDA-based formulation, the piezoresistive properties of the nanocharged resin have been evaluated using a concentration of 0.3 wt.% (by weight) of MWCNTs, for which the nanocomposite is just beyond the electrical percolation threshold (EPT). The sensitivity of a piezoresistive nanomaterial, which can be expressed in terms of gauge factor (G.F.) and calculated as the ratio of relative change in electrical resistance R, to the mechanical applied strain ε (i.e., G.F. = ΔR/εR_0_), has been found to be 0.43 [15]. The self-sensing performance has been analyzed also for the formulation based on the DGEBA precursor containing the same kind of multi-wall carbon nanotubes (MWCNTs) [17]. For the DGEBA-based formulation, the gauge factor has been evaluated for different percentages (wt./wt.) of MWCNTs [17]. It has been found that the G.F. decreases with increasing the MWCNT percentage [17,30]. For both formulations, damages involving nanometric domains seem directly correlated to the changes in the electrical resistance and result easily detectable in a non-destructive way through electrical measurements [15,17]. During fatigue tests, damages can be detected by the presence of a residual resistivity, which is dependent on the amount of plastic strain accumulated in the sample [15,17]. The sensor property of the DGEBA-based formulation has also been analyzed in the form of a coating on carbon fiber reinforced panels (CFRPs). Piezoresistive and mechanical properties of the CFRPs coated with DGEBA-based resin with 0.1 wt.% of MWCNT have been evaluated for the panel subjected to axial tension [16]. Electrical properties and mechanical strain correlations of coated panels highlighted the feasibility of the developed strategy for integrating high sensitivity and effective real-time structural health monitoring in aeronautical panels. Self-responsive CFRPs can perform multiple functions acting as sensor and structural-health monitoring element [31].

A suitable choice of the epoxy precursor/hardener and CNT nature and concentration allows preparing self-sensing resins with a satisfactory sensitivity factor and a high mechanical performance, conditions that make the strain sensors reliable in the normal operating temperature range of structural materials such as aeronautics, automotive field and civil engineering. Considering the cost and the mechanical requirements, it is possible to opt for one choice rather than another.

## 2. Materials and Methods

The epoxy matrix has been prepared by mixing an epoxy precursor, tetraglycidyl methylene dianiline (TGMDA) (Sigma Aldrich St.Louis, MI, USA) with an epoxy reactive diluent, 1, 4-butanediol diglycidyl ether (BDE) (Sigma Aldrich St.Louis, MI, USA). The 4,4-diaminodiphenyl sulfone (DDS) (Sigma Aldrich St.Louis, MI, USA) has been used as curing agent of the epoxy mixture TGMDA/BDE (concentration by weight of 80:20%, respectively). After a mixing (obtained at 120 °C for 1 h), the mixture has been cured at 125 °C for 1 h followed by a temperature cycle at 200 °C for 3 h. A homogeneous dispersion of the carbon nanotubes (MWCNT), (3100 Grade Nanocyl S.A., Sambreville, Belgio), has been obtained by using an ultra-sonication process for 20 min (Hielscher model UP200S-24 kHz high power ultrasonic probe) (Hielscher Ultrasonics Teltow Germany). The epoxy composite systems based on diglycidyl ether of bisphenol A (DGEBA) (Sigma Aldrich St.Louis, MI, USA) and DDS have been prepared with a similar procedure but with a different curing cycle: 150 °C for 1 h followed by 3 h at 220 °C. Two different curing cycles have been chosen in order to have a curing degree of 97%. The epoxy systems have been characterized by different experimental techniques summarized in Table 1.

In order to investigate the sensing properties behavior of the nanocomposites, axial response strength measurements were performed using a Dual Column Tabletop Testing Systems (INSTRON, series 5967) set with a cross head speed of 1 mm/min. The corresponding force was measured by the machine load cell and converted to axial stress (σ), whereas mechanical strain (ε) was calculated as the machine crosshead displacement normalized by the gage length of the test specimen. In order to exclude possible slipping during the displacement, the local deformation was detected by means of a conventional strain gage (RS 5 mm Wire Lead Strain, gauge factor 2.1) bonded to one side of the specimen and having a gauge resistance of 120 Ω constantly measured with a precision multimeter HP 34401A. Copper electrodes were fixed on the sample surface using silver paint (Silver Conductive Paint, resistivity of 0.001 Ω cm) thus ensuring good ohmic contact between the parts for the measurement of the resistance, R, of the samples using the two-probe method with a Multimeter Keithley 6517A configured in the double function of voltage generator and ammeter. Contact resistance was neglected since the measured electrical resistance for all specimens was in the order of several kΩ.

## 3. Results and Discussion

### 3.1. Morphological and Electrical Investigation

In order to obtain the highest possible values of the gauge factor (obtainable with the formulated systems) a range of carbon nanotube concentrations between 0.05% and 0.5% was selected.

Figure 1 shows the HRTEM images of the epoxy nanocomposite based on the TGMDA precursor. MWCNTs have been dispersed in the resin at different concentrations: 0.05, 0.10, 0.30, and 0.50 wt.%.

From the TEM images, it is possible to observe that the first electrical percolation paths of MWCNTs can be observed starting from a concentration of 0.30 wt.% of MWCNTs. For the weight percentage of 0.50 wt.% of MWCNTs, in all the analyzed zones (not all shown here), MWCNTs appear to be well interconnected. They form electrically conductive paths in all the fracture surfaces of the sample. This morphology is perfectly in line with the value of the electrical conductivity observed in Figure 2 where the electrical conductivity (σ), as a function of the filler concentration (wt.%), is reported for the TGMDA based formulation.

The nanocomposite undergoes an insulator-to-conductor transition when the conductive filler loading is gradually increased up to the achievement of a critical concentration, the so-called electrical percolation threshold (i.e., EPT). In this case the EPT is below the 0.3%, showing a good agreement with the observations drawn from the HRTEM images of Figure 1.

Figure 3 shows the HR-SEM images of the epoxy nanocomposite based on the DGEBA precursor. MWCNTs have been dispersed in the resin at concentrations of 0.05, 0.10, 0.30 and 0.50 wt.%.

Regarding the DGEBA-based formulation, the morphological results highlight that, differently from that based on the TGMDA formulation, both the samples containing incorporated 0.10 and 0.30 wt.% of MWCNTs manifest the presence of CNTs well interconnected among different zones, originating conductive paths through the samples. This result perfectly agrees with the EPT curve of this system, shown in Figure 4, which manifest a lower EPT with respect to the one based on the TGMDA formulation of Figure 2. In fact, Figure 4 highlights that the system is just after the EPT for a concentration of MWCNTs corresponding to 0.1% by weight.

Figure 5 shows a comparison of the bulk conductivity for the two different polymeric resins reinforced with different concentrations of MWCNTs. It is evident that, for the DGEBA based system, it is possible to obtain higher values of electrical conductivity for lower amount of MWCNTs.

Although the precursors adopted for the two formulations are epoxy precursors, and they have been hardened with the same hardener (in stoichiometric amount), the functionality of the epoxy precursor strongly affects the morphological and electrical properties. From the electrical point of view, the difference is much more pronounced especially around the electrical percolation threshold. The two systems differ in both the electrical conductivity values and the electric percolation thresholds. Following a classical approach widely discussed in the literature [34,35,36], it is reasonable to maintain that the electrical conduction in the analyzed composites is attributable to the tunneling effect.

In particular, the tunneling effect is a function of the filler dispersion, its aspect ratio and the different interfacial contact area [12]. Considering that for both formulations, the filler and the process adopted for the dispersion is the same, it is very likely that the observed difference can be ascribed to the distance among the nanoparticles, with a very relevant effect around the EPT. In fact, at the concentration 0.1% wt/wt of the filler, the two systems differ in the electrical conductivity value by approximately 11 orders of magnitude. The study of the mechanical-dynamic properties, discussed in the next section, of the two different developed systems can help to better understand this different behavior.

### 3.2. Dynamic Mechanical Properties

Curve profiles of storage modulus and Tan δ vs. temperature for the two analyzed formulations are shown in Figure 6a,b. The comparison between the values of the storage modulus is shown in Figure 6a, whereas that related to the values of loss factor (tan δ) is presented in Figure 6b.

Modulus values of the TGMDA-based formulations are always higher than those related to the DGEBA-based formulation.

However, for both formulations, the storage modulus is higher than 2000 MPa in the temperature range from −60 to 70 °C, fully satisfying the requirements fixed for epoxy resins intended for structural composites in the aeronautic and automotive transport sectors. Going from −70 to 210 °C, modulus values slowly and progressively decrease before the principal drop, between 220 and 270 °C, due to the glass transition temperature (i.e., T_g_), which exactly falls in the interval from 200 to 300 °C (see Figure 6b). Figure 6b shows a shift in the temperature value of the main peak for the TGMDA-based formulation towards higher values of temperature. It is well known from polymer theory that the relaxation process in polymers and especially in composites are complex processes which stem from different mechanisms [37]. Among various theoretical models, which have been proposed to describe these complex effects, the assumption that the recorded process consisting of discrete elementary relaxation processes, results in a realistic and practical approach. These processes can be analyzed by the study of the glass transition temperature of the polymer. The glass transition process may be considered as being controlled by the intrinsic flexibility of the chain segments of the polymer (e.g., between two nodes of the network) or as a function of the free volume available within the polymer. Conformational changes can only occur when there is enough free volume to allow chain segment movements. The free volume is assumed to be present throughout the polymer. Moreover, according to the approach of the flexibility of the chain segments, the variation of chemical structure also involves an intermolecular contribution. Thus, the glass transition can be influenced by both the intra- and inter-molecular contributions. Other factors, such as kinetic and curing degree (depending on the adopted heating procedure) and the incorporation of fillers and/or small molecules, such as water molecules, also condition the profile of dynamic mechanical graphs. The height and width of the peaks in tan δ spectra can be evaluated to draw information on the relaxation mechanisms due to the different curing procedures. The height of the main peak for the DGEBA system is 0.8, while for TGMDA system is 0.56. This implies that the TGMDA system exhibits a more elastic behavior than the bi-functional system. The significant change in the width of the peak also suggests a broader distribution of relaxation times, presumably due to a more gradual activation of the segment mobility. The higher value of the Tan δ peak for the TGMDA system suggests that to activate the segment mobility, higher values of temperature are required with respect to the DGEBA based system, and therefore a more restricted mobility at lower values of temperature. This observation perfectly agrees with the higher values of storage modulus detected for the TGMDA system in the range of temperature lower than 270 °C. Table 1 shows the different chemical formula of the two different epoxy precursors, their functionality, and the supplier. The structure of the chemical formulation can help to understand the different behavior of the formulated nanocomposites.

As shown in Table 2, the epoxy precursor of the TGMDA precursor is characterized by four functionalities, two more than the formulation based on the DGEBA precursor. The higher number of functionalities allows obtaining a higher crosslinking density with less mobility of chain segments. The crosslinking density can be described by Equation (1) [38]
(1)$$\rho =\frac{E^{\prime }}{3RT}$$
where *R* is the universal gas constant (8.314 J K^−1^ mol^−1^); *T* is equal to T_g_ +50 K, *E*′ is the storage modulus at temperature of T_g_ + 50 K. The crosslinking density can be expressed as *Mc*, which is the molecular weight between entanglements or crosslink sites. The value of the storage modulus in the rubbery plateau region is inversely proportional to the chain length between entanglements, M_c_, and is given by Equation (2) [38,39]:(2)$$M_{c}\propto \frac{1}{E^{\prime }}$$

Table 3 shows the values of *E*′ at the temperature T_g_ + 50° K for the two systems. For the TGMDA sample a value of *E*′ approximately three times higher than the sample DGEBA is detected, most likely due to the higher crosslinking sites, determined by the higher functionality number.

A greater crosslinking density results in a greater rigidity of the TGMDA based system with a consequent higher glass transition temperature (see Figure 6). During the crosslinking process, the conductive paths formed by the nanotubes can be interrupted by the ever-increasing formation of crosslinking points that act as an obstacle to the distribution of the nanotubes. This effect is all the more accentuated as the net of the matrix becomes denser. For this reason, the different crosslinking density of the two systems affects the distribution of carbon nanotubes in the two considered matrices and influences the range corresponding to the electric percolation threshold. Figure 7 shows the DMA results related to the two systems containing dispersed CNTs. In particular, the comparison between unfilled and filled epoxy formulations with a percentage of 0.5%wt/wt is shown.

Variations of the glass transition temperature value are due to the balancing of opposite phenomena. Generally, for the same curing condition, the presence of the nanofiller in the polymeric matrix tends to decrease the curing degree with respect to the unfilled matrix [4]. This effect is poorly reflected on the T_g_ value because the tendency to a decrease in T_g_ value is counterbalanced by the restricted mobility of the polymer chains in the vicinity of high-aspect-ratio carbon nanotubes [40,41]. These opposite effects result in an almost unchanged value of T_g_ for the matrix and the nanocomposite [41]. This consideration agrees well with the results of Figure 7c, where almost the same value of T_g_ is obtained for the unfilled resin and the nanocomposite DGEBA based system. The lack of a change in the value of T_g_, together with the increase in the storage modulus of the filled system, in the range of temperatures higher than the environment temperature (25 °C) and also above the T_g_, leads to the conclusion that the filler reinforcement effect occurs without interruption of crosslinking density, or at least that it is not so relevant as to affect the value of the glass transition temperature. Contrariwise, the filled TGMDA system presents the storage modulus values similar to that of the unfilled resin with a clear decrease in the modulus in the rubber phase (see inset Figure 8b). Besides this trend, the filled system shows a decrease in the height of the main peak of tan d and the presence of a new crosslinking peak at a lower temperature (215 °C) (Figure 7d). From DMA behavior, it is clear that in the TGMDA system, the crosslinking interruption phenomenon is relevant with respect to that obtained in the DGEBA system.

The different mechanical behavior obtained for the two systems, due to the introduction of the filler, explains the different EPT values found in Figure 2 and Figure 4. The system, which has a precursor with two functionalities, with a lower crosslinking density than the TGMDA system, is characterized by a higher free volume value. The greater free volume allows a better distribution of the filler without no relevant limited interruption of the conductive paths for the effect of the formation of the crosslinking reactions. In the system with the tetra-functional precursor, the crosslinking reactions interrupt more effectively the conductive network, causing a reduction in the electrical conductivity of the filled sample. Only with an increase in the concentration of the filler, it is possible to obtain electrically conductive paths comparable to that of the bi-functional system.

### 3.3. Correlation between Epoxy Matrix Structure and Sensing Properties

In light of this interpretation, it is possible to explain more clearly the differences between the considered systems in relation to the gauge factor values obtained in previous published papers [15,17]. Figure 8 summarizes some results obtained from Vertuccio et al. in which the TGMDA and DGEBA systems are considered. With the same concentration of carbon nanotubes, 0.5% wt/wt, the gauge factor is very similar (0.43 and 0.63 for TGMDA and DGEBA systems, respectively). In the case of the TGMDA system, the value of 0.43 has been detected for a value of CNT concentration (0.5% wt/wt) close to EPT, while the value of 0.63, for the same amount of CNTs, was obtained for a concentration very far from the EPT value. In general, the sensitivity of the sensor can be improved by decreasing the filler concentration [17,30]. The closer the filler concentration to the percolation threshold value, the greater the sensitivity of the sensor. This is true as long as the correspondence between the change in electrical resistance and the applied strain is linear or at least almost linear. A linear trend ensures stable sensor behavior under load cycles. If the concentration is very close to the EPT value, the correspondence just mentioned above behaves like an exponential [11,42]. This may determine the definitive breaking of the contacts between the particles, also detectable from the stress/strain curve, not ensuring the recovering of the resistance in zero load condition during load cycles. In the case of the bi-functional system, the gauge factor can be improved, without having the above drawbacks, up to a GF value of 1 detected for a concentration of CNTs corresponding to 0.1% wt/wt. The TGMDA system, being characterized by a higher EPT range than that of the bi-functional system, does not allow consistent improvements in the G.F. for low concentration of CNTs. Considering that the chosen matrices only differ for the functionality, which affects the crosslinking density, it is very likely to assume that the different electrical behavior, as the mechanical properties, is a direct consequence of the different functionality of the epoxy precursor. In fact, the functionality controls the crosslinking density, affecting the conductive network in the matrix and therefore conditioning the sensor properties.

## 4. Conclusions

In the present work, the effect of the structure of matrix of resins containing CNTs, suitable for applications in automotive and aeronautic sectors, on mechanical and sensing properties has been investigated. The different functionality of the epoxy precursors affects the crosslinking density of the resulting material determining the different behavior of the mechanical, electrical and sensing properties. The use of a tetra-functional epoxy precursor, rather than the bi-functional one causes different effects, among which is a very effective interruption of the electrically conductive network of CNTs with a consequent increase in the EPT value. Furthermore, the presence of CNTs in the precursor with higher functionality determines a two-phase composition, with domains characterized by higher chain mobility, detected by the splitting in two peaks of the transition regions in tan d curve. The functionality of the precursor also affects the sensitivity of strain sensors.

The choice of one precursor rather than the other must be carried out according to the performance required for the specific application. In the criteria for the choice, the structure of the components and their polymerization process also has to be taken into account.

## Figures and Tables

**Figure 1 nanomaterials-11-00833-f001:**
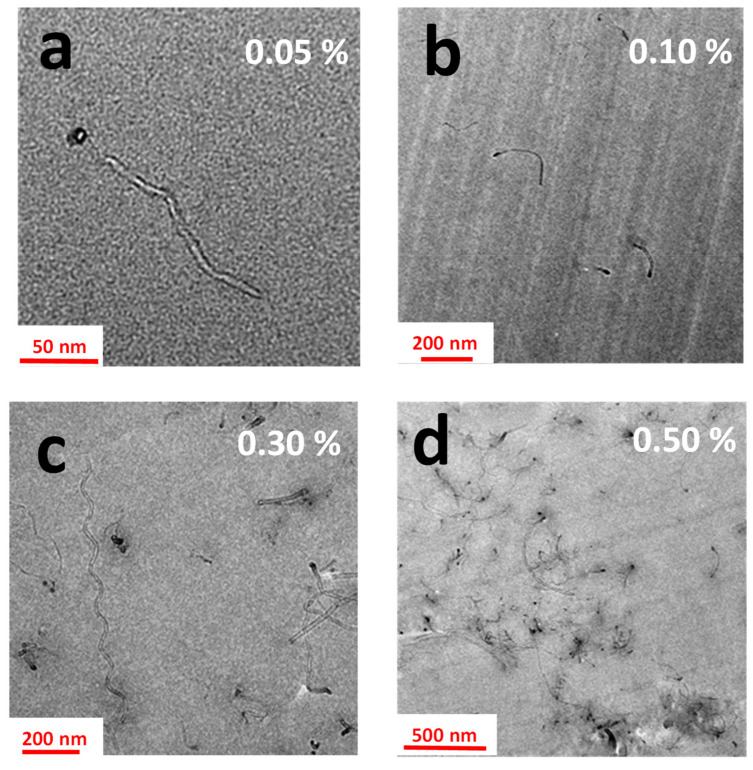
HRTEM images of the tetraglycidyl methylene dianiline (TGMDA) formulation containing embedded different concentrations of multi-wall carbon nanotubes (MWCNTs): (**a**) 0.05 wt.%; (**b**) 0.10 wt.%; (**c**) 0.30 wt.%; (**d**) 0.50 wt.%.

**Figure 2 nanomaterials-11-00833-f002:**
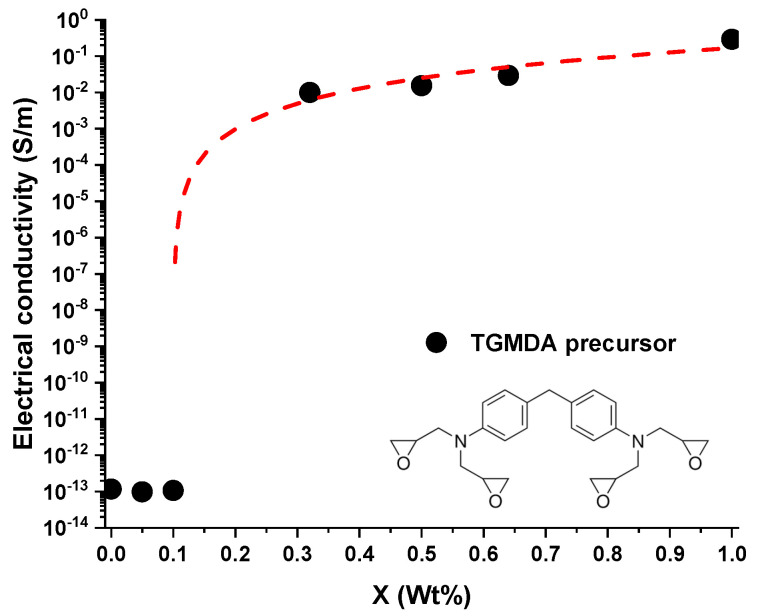
DC volume electrical conductivity (s) of the TGMDA-based formulation versus MWCNT weight percentage.

**Figure 3 nanomaterials-11-00833-f003:**
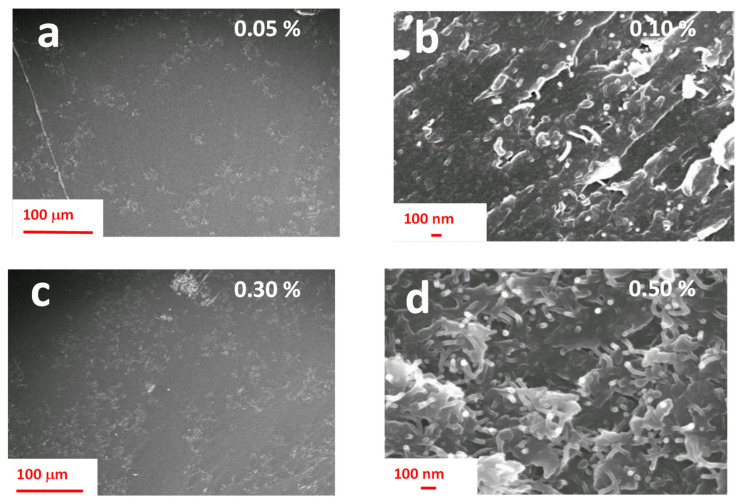
HR-SEM images of the diglycidyl ether of bisphenol A (DGEBA) formulation containing embedded different concentrations of MWCNTs: (**a**) 0.05 wt.%; (**b**) 0.10 wt.%; (**c**) 0.30 wt.%; (**d**) 0.50 wt.%.

**Figure 4 nanomaterials-11-00833-f004:**
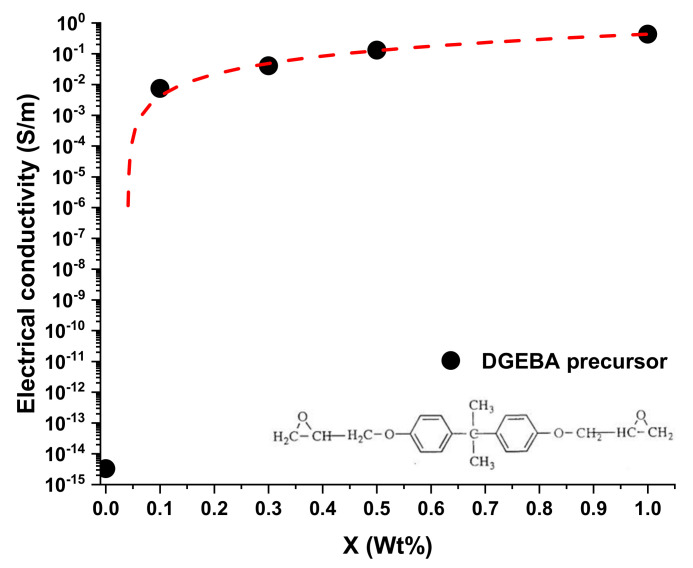
DC volume electrical conductivity (s) of the DGEBA-based formulation versus MWCNT weight percentage.

**Figure 5 nanomaterials-11-00833-f005:**
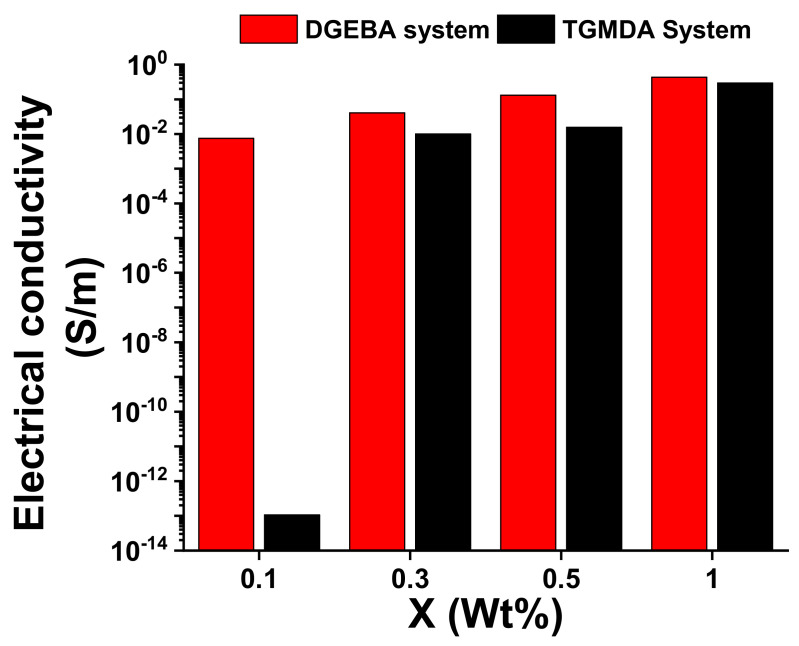
Comparison of the bulk conductivity for the two different polymeric resins reinforced with different concentrations of MWCNTs.

**Figure 6 nanomaterials-11-00833-f006:**
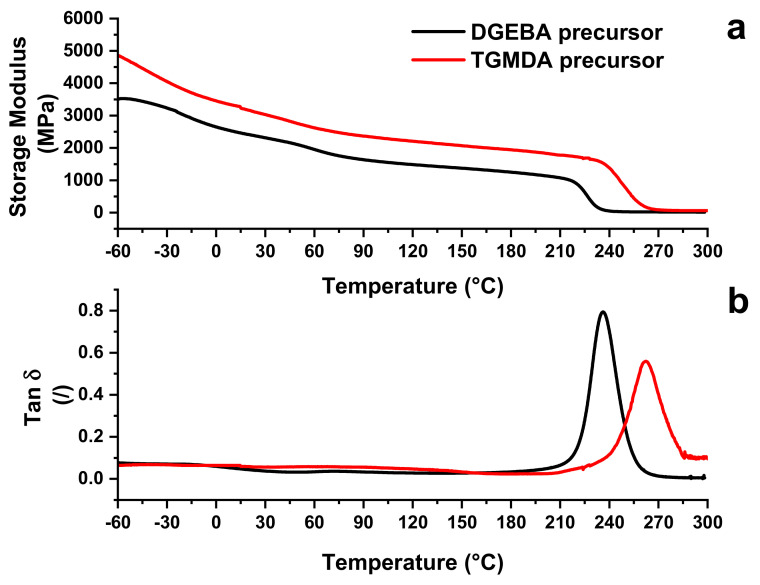
(**a**) DMA relating to TGMDA and DGEBA systems: (**a**) storage modulus; (**b**) Tan δ.

**Figure 7 nanomaterials-11-00833-f007:**
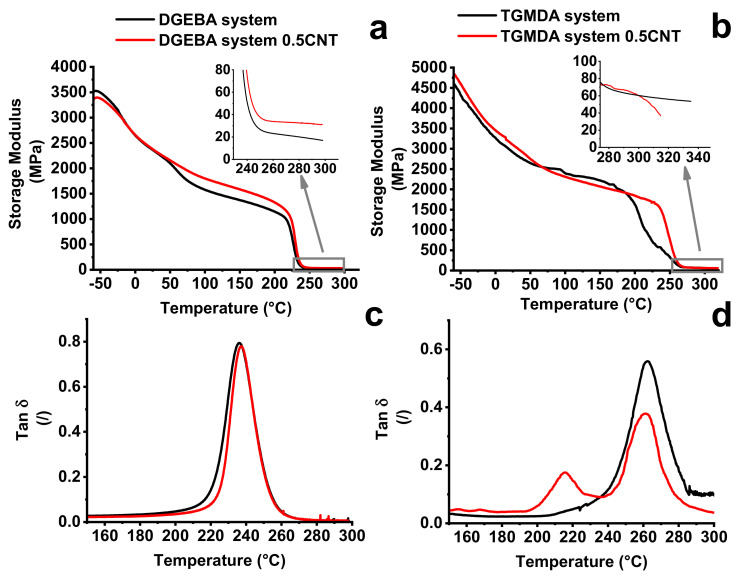
DMA results of the two systems: (**a**) curves of storage modulus of the unfilled and filled (0.50% wt/wt of CNTs) DGEBA system; (**b**) curves of storage modulus of the unfilled and filled (0.50% wt/wt of CNTs) TGMDA system; (**c**) Tan δ of the unfilled and filled DGEBA system; (**d**) Tan δ of the unfilled and filled TGMDA system.

**Figure 8 nanomaterials-11-00833-f008:**
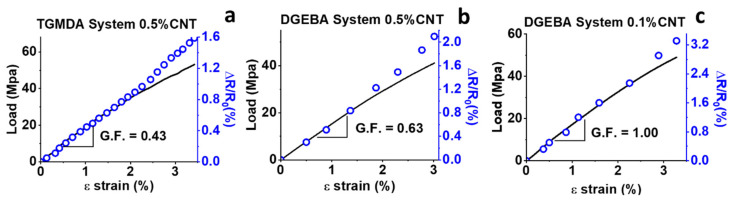
Mechanical response (σ, left vertical axis) and resistance change ratio (ΔR/R_0_, right vertical axis) observed in tensile stress as a function of the axial strain (ε) of: (**a**) TGMDA system with 0.5% CNTs; (**b**) DGEBA system with 0.5% CNTs; (**c**) DGEBA system with 0.1% CNTs.

**Table 1 nanomaterials-11-00833-t001:** Characterization methods.

**Dynamic Mechanical Analysis**	**Specific**
Sample dimension	4 × 10 × 35 mm^3^ (thickness, width and length, respectively)
Configuration	Dual Cantilever
Displacement amplitude	0.001
Frequency operating condition	1 Hz
Temperature operating condition	from −60 °C to 300 °C
Scanning rate	3 °C/min^–1^
Device	TA instrument-DMA 2980, New Castle, DE, USA
**Electrical measurement**	**Specific**
Sample dimension	2 mm × 50 mm (thickness and diameter, respectively)
Configuration	2-wire method according to [12]
Contact	coating deposition by silver paint
Device	Electrometer Keithley 6517A(Keithley Instruments, Cleveland, OH, USA)
**Scanning Electron Microscopy**	**Specific**
Procedure	According to ref. [32]
Device	JSM-6700F, (JEOL Akishima, Japan)
**Transmission Electron Microscopy**	**Specific**
Procedure	According to ref. [33]
Device	JEOL model JEM-1400 Plus (JEOL Akishima, Japan)

**Table 2 nanomaterials-11-00833-t002:** Chemical formulas of the two epoxy precursors, functionality and supplier.

Precursor	Formula	Supplier	Functional Group
**DGEBA**	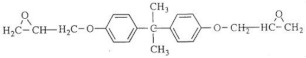	Sigma Aldrich	2
**TGMDA**	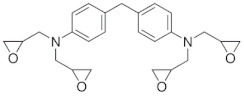	Sigma Aldrich	4

**Table 3 nanomaterials-11-00833-t003:** Crosslink parameters.

Sample	*E*’ (T_g_ + 50° K) MPa	T_g_ (°C)	ρ (mol/cm^3^)
DGEBA	19.02	236	1.50 × 10^3^
T20BD	57.52	262	4.30 × 10^3^

## Data Availability

Data sharing not applicable.

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
