# Peer review of "Self-Sensing Nanocomposites for Structural Applications: Choice Criteria"

_nanomaterials, 2021, doi:10.3390/nano11040833_

Round 1

Reviewer 1 Report

(1)In 2. Materials and Methods section, please provide more information of how to measure sensing property.

(2)In this work, the authors fix the concentrations of MWCNTs as 0.05wt% to 0.5wt%, please address the reason. And how about the effect of more than 0.5wt%.

Reviewer 2 Report

This manuscript has focused on self-sensing nano-composite of epoxy resins containing multi-wall carbon nanotubes.

The article is certainly of great interest, especially from the point of view of practical application. This is a typical high-quality (state-of-the-art) study of MWCNT nanocomposite and epoxy resin. All the principal aspects of the preparation of the nanocomposite are well described in the work, including a homogeneous dispersion of the carbon nanotubes and HRTEM study of the resulting nanocomposite. The structure, mechanical properties and the dependence of the mechanical properties of the nanocomposite on the structure are investigated.

The novelty of the article is as follows. Two different epoxy precursors were studied: tetrafunctional tetraglycidyl methylene diane lane (TGMDA) and bifunctional Bisphenol A diglycidyl ether (DGEBA). The advantage of using TGMDA in conditions where high values in the glass transition temperature and storage modulus are required is shown.

The disadvantage of the article is that the problems of interaction of a single nanotube with an epoxy resin are not investigated, as well as an effect of epoxy crystallization in the presence of carbon nanotubes, etc. (for example, this problem was raised in the work of Russian Chemical Reviews 79 (11) 945- 979 (2010)).

However, this remark (rather a wish) is not an obstacle to the publication of this article. The article is well written, the text is clear and easy to read, the conclusions are consistent with the presented evidence and arguments.
